

# Life expectancy of cats in Britain: moggies and mollies live longer

Fernando Mata

Center for Research and Development in Agrifood Systems and Sustainability, Instituto Politécnico de Viana do Castelo, Viana do Castelo, Portugal

## ABSTRACT

The domestic cat (*Felis catus*) has been a popular companion animal for about 12,000 years, initially valued for rodent control before evolving into pets that provide affection and companionship. Unlike dogs, cats were not selectively bred for specific roles until the late 1800s, resulting in breeds defined primarily by appearance, which sometimes leads to genetic disorders. Modern animal welfare concerns emphasize longevity and health, prompting research into factors affecting cat lifespans, including sex, reproductive status, and breed. This study aims to expand on previous UK research by analyzing these interactions and highlighting the potential negative impacts of pure breeding on cat health. Data from 7,708 cats receiving veterinary care in the UK during 2019 were analyzed, focusing on reproductive status, breeding status, age at death, and sex. Data were analyzed using ANOVA and Cox proportional hazards models to assess survival differences. The overall mean lifespan of cats in the UK is 11.83 years. Analysis indicates that male cats live shorter lives than females, attributed to higher trauma rates and health issues among males. Neutered/spayed cats generally exhibit longer lifespans compared to entire cats. Tom cats have the shortest lifespan, while spayed females (mollies) live the longest. Moggies tend to outlive both pure and cross-bred cats, suggesting that genetic diversity may contribute to greater longevity. The results of this study emphasize the influence of sex, reproductive status, and cat type on feline lifespan, highlighting the need for targeted health interventions, particularly for male cats. The findings underscore the complex interplay of genetic and environmental factors in determining the health and longevity of domestic cats. This research not only contributes to existing knowledge but also advocates for the consideration of these variables in future studies and veterinary practices.

## INTRODUCTION

The domestic cat (*Felis catus*) is one of the most popular companion animals worldwide. Its domestication began approximately 12,000 years ago in the Fertile Crescent and the Levante, with the process being completed in Egypt 6,000 years ago (*Nilson et al., 2022*). Cats were primarily kept in households for their hunting abilities and rodent control (*Crowley, Cecchetti & McDonald, 2020*). They were tolerated in homes due to their pest control skills, and some authors suggest a process of self-domestication in a commensal

Corresponding author
Fernando Mata,
fernandomata@ipvc.pt

relationship (*Hu et al., 2014*). The relationship and tolerance between humans and cats evolved naturally due to the benefit for both humans and cats. While cats benefited from the availability of food in a humanised environment, humans benefited from the presence of cats as these could provide a way to control pests. The role of companion animals developed later in the history of the human-cat relationship (*Crowley, Cecchetti & McDonald, 2020*).

As household pets, cats provide a range of benefits, including enjoyment, affection, companionship, and even health advantages (*Amiot, Bastian & Martens, 2016*). However, these roles have not led to the specialization observed in dogs, such as herding, guarding, and game hunting. Consequently, differentiation through selection was not evident until the late 1800s, when Victorian Britain began breeding cats for their appearance, marking the evolution of the cat fancy (*The British Newspaper Archive, 2013*). The National Cat Club, the first cat club, played a key role in promoting the cat fancy, and together with a second club, it became the Governing Council of the Cat Fancy in Great Britain, established in 1910 and recognized as the oldest fancy cat association.

Breeds evolved based on appearance rather than specialization, leading to the fixation of deleterious genes resulting from mutations, such as those seen in brachycephalic cats (*e.g.*, Persian and Burmese) (*Plitman et al., 2019*), the furless Sphynx (*Åhman & Bergström, 2009*), the cartilage disorder causing the Scottish Fold's distinctive appearance (*Gandolfi et al., 2016*), and the shortened limbs seen in the Munchkin (*Lyons et al., 2019*).

Modern animal welfare science has highlighted these genetic issues, and the longevity of pets has gained importance as a concern for both pet owners and veterinary professionals (*e.g.*, *Farstad, 2018*; *Grandin & Deesing, 2022*; *Morel et al., 2024*; *Sand et al., 2017*). Data on mortality and life expectancy are crucial for understanding the health and welfare of companion cat populations. Predicted life expectancy can reveal shifts in overall health and welfare, and comparing lifespans across different cat subgroups can help identify those at higher risk for health issues or shorter lifespans (*O'Neill et al., 2015*). Previous research has highlighted various biological (*e.g.*, *Kent et al., 2022*; *Mata & Bhuller, 2022*), environmental (*e.g.*, *Kent et al., 2022*; *Morel et al., 2024*), and behavioral (*Mata, 2015*; *Montoya et al., 2023*; *Teng et al., 2024*) factors that affect cat longevity, including reproductive status, sex, breed, and genetic factors (*Egenvall et al., 2009*).

While referring to cats' breeding, the term *domestic cat* is frequently used to describe non-pedigreed cats, commonly known as moggies in the UK or domestics elsewhere. Unlike pedigreed breeds, which are selectively bred for specific traits, domestic cats are the result of natural genetic diversity and lack standardized physical or behavioral characteristics.

Several methods can be used to report overall mortality: average (mean or median) age at death, life table analysis, and survival analysis. Life tables allow for estimating life expectancy and the associated probability of death at specific age groups. Survival analysis, using Kaplan-Meier and Cox regression methods, complements life table analysis by enabling comparisons between groups with different characteristics and risk factors in single models (*Abd ElHafeez et al., 2021*).

Previous studies, including those by *O'Neill et al. (2014)* and recently *Teng et al. (2024)* in the UK, have constructed life tables, while *Kent et al. (2022)* in California, USA, calculated lifespans for different cat groups, and *Montoya et al. (2023)* generated life tables based on clinical records from Banfield Pet Hospitals in the USA.

This study aims to further investigate the lifespan of cats in the UK, extending the research conducted by *Teng et al. (2024)* with a survival analysis focusing on how cat genetic type, reproductive status, and sex interact to influence lifespan. By analyzing the interactions between these variables, this study seeks to provide a more comprehensive understanding of their contributions to feline longevity. Particularly concerning breeding effects, we aim to demonstrate that pure breeding may have negative implications for the health and welfare of cats, as previously shown in dogs by *Mata & Mata (2023)*.

## MATERIALS AND METHODS

The data used in this study were publicly available under a Creative Commons Attribution 4.0 license (CC BY 4.0) and were obtained from the Royal Veterinary College, University of London's repository on FigShare (*O'Neill & Teng, 2024*). The sample consists of all cats receiving primary veterinary care in clinics affiliated with the VetCompass™ Program in the UK during 2019, meaning they had at least one electronic patient record in that year (*Teng et al., 2024*). The dataset included variables for 'reproductive status' (entire and neutered/spayed), breeding status or 'cat type' (identifying pure breeds and cross-breeds as cats with only one pure breed ancestor, and moggies as all types of domestic cats without any type of purebred ancestry), 'age at death' in years, and 'sex' (male and female). Spayed females will, from now on, be referred to as mollies.

Initially, the dataset contained $N = 7,935$ entries. However, after cleaning the data by removing unknown breed entries, unknown reproductive statuses, and outliers, the final sample included $N = 7,708$ cats. Detailed figures of observations in the different categories can be found in Table 1. Outliers were detected following the transformation of the data to a standard normal distribution, characterized by a mean of zero and a standard deviation of one, by identifying values with z-scores exceeding 3 or falling below −3 standard deviations.

An ANOVA with a least significant difference (LSD) *post-hoc* test was used as a full factorial general linear model to compare mean differences between the 'cat types,' 'reproductive status,' and 'sex,' along with their respective interactions. The LSD test was chosen as the homogeneity of variances was verified, and no more than three levels were compared for each factor analysed.

A one-way ANOVA was also performed using the factor joining together sex and reproductive status, with the levels 'entire female', spayed female', 'neutered male', 'entire male'. Tukey's honest significant difference (HSD) test was used as the *post-hoc* as there were four levels in the factor analysed.

The prerequisites of the ANOVA—namely, homogeneity of variances and normal distribution of residuals—were assessed and confirmed *via* Levene's test and *via* Kolmogory-Smirnov test, respectively.

**Table 1 Crosstab with the overall number of observations and per category sex × reproductive status × breeding status.**

| Breeding status | Sex | Reproductive status | | Total |
|---|---|---|---|---|
| | | Entire | Neutered/Spayed | |
| Cross bred | Female | 42 | 93 | 135 |
| | Male | 27 | 91 | 118 |
| | Total | 69 | 184 | 253 |
| Moggies | Female | 896 | 2,344 | 3,240 |
| | Male | 887 | 2,524 | 3,411 |
| | Total | 1,783 | 4,868 | 6,651 |
| Pure breed | Female | 132 | 254 | 386 |
| | Male | 118 | 300 | 418 |
| | Total | 250 | 554 | 804 |
| Total | Female | 1,070 | 2,691 | 3,870 |
| | Male | 1,032 | 2,915 | 3,947 |
| | Total | 2,102 | 5,606 | 7,708 |

The data were analyzed using a Cox proportional hazards model to assess survival, with 'age at death' as the time-to-event variable and 'cat type' (moggy, crossbred, or purebred), 'reproductive status' (entire or neutered/spayed), and 'sex' as the factors of interest. No data were censored, and the event was defined as the 'age at death' (in years). The model was evaluated using the −2 Log likelihood test, while the parameters were assessed with the Wald test. Cumulative survival plots were generated. The prerequisites of linearity and proportional hazards assumptions are met.

To investigate the interaction between 'reproductive status' and 'sex,' a new variable combining both factors was created, resulting in four levels: 'mollies,' 'female entire,' 'male neutered,' and 'male entire.' The data were entered into a Kaplan-Meier model tested with a log-rank (Mantel-Cox) test to calculate differences between lifespan means and medians. A survival plot was also produced.

Data were first imported into a spreadsheet (Microsoft Excel for Microsoft 365 MSO, version 2204 Build 16.0.15128.20240, 64-bit) for cleaning. Descriptive statistics were also computed using this software. The Cox proportional hazards snd the Kaplan-Meier models were fit using the software package IBM SPSS Statistics (Version 29.0.2.0 (20); IBM Corp., Armonk, NY, USA). The level of significance was set to $p < 0.05$ in all tests.

## RESULTS

The overall mean lifespan for the cats in the data set analysed was 10.61 years with a 95% confidence interval of [10.30–10.91].

The ANOVA test yielded a significant result, indicating significant differences in all main effects and the interaction between the 'reproductive status' and the 'cat type'. Details of the ANOVA results are displayed in Table 2. The results of the *post hoc* tests showing the mean values found significant, are shown in Table 3.

**Table 2 Full factorial analysis of variance (ANOVA) used to test significant differences in lifespan between 'cat type' (moggy, cross-bred, cross breed), 'reproductive status' (entire, neutered/spayed), 'sex' (male, female), and interactions.**

| Source | Sum of squares | df | Mean square | F | p-value |
|---|---|---|---|---|---|
| Model | 1,091,907.80[a] | 12 | 90,992.32 | 2,853.95 | <0.001 |
| Sex | 793.42 | 1 | 793.42 | 24.89 | <0.001 |
| Reproductive status | 2,688.81 | 1 | 2,688.81 | 84.33 | <0.001 |
| Cat type | 1,800.98 | 2 | 900.49 | 28.24 | <0.001 |
| Sex × Reproductive status | 18.46 | 1 | 18.46 | 0.58 | 0.447 |
| Sex × Cat type | 17.55 | 2 | 8.78 | 0.28 | 0.759 |
| Reproductive s × Breeding status | 426.12 | 2 | 213.06 | 6.68 | 0.001 |
| Sex × Reproductive s × Cat type | 90.62 | 2 | 45.31 | 1.42 | 0.242 |
| Error | 245,370.91 | 7,696 | 31.88 | | |
| Total | 1,337,278.71 | 7,708 | | | |

**Note:**
[a] $R^2 = 0.817$, Adjusted $R^2 = 0.816$; S, status.

**Table 3 Mean lifespan (years), and respective 95% confidence intervals (CI) for the different levels of factors, and *post-hoc* tests to the factors and interactions found significant in the ANOVA test.**

| Factor | Level 1 | Level 2 | Level 3 | Level 4 | Level 5 | Level 6 |
|---|---|---|---|---|---|---|
| Sex | Male | Female | ——— | ——— | ——— | ——- |
| Mean | 11.17[a] | 12.52[b] | ——— | ——— | ——— | ——— |
| 95% CI | [10.99–11.35] | [12.33–12.70] | | | | |
| Cat type | Moggy | Cross-bred | Pure breed | ——— | ——— | ——— |
| Mean | 12.01[b] | 11.03[a] | 10.56[a] | ——— | ——— | ——— |
| 95% CI | [11.87–12.15] | [10.32–11.74] | [10.16–10.96] | | | |
| Rep. St. | Entire | N/S | ——— | ——— | ——— | ——— |
| Mean | 11.83[a] | 12.45[b] | ——— | ——— | ——— | ——— |
| 95% CI | [9.91–10.40] | [12.30–12.60] | ——— | ——— | ——— | ——— |
| Cat type × Rep. St. | E. P. B. | E. C.-B. | E. Moggie | N/S P.B. | N/S C.-B. | N/S P.B. |
| Mean | 8.02[a] | 9.03[b] | 10.49[b] | 11.78[c] | 11.72[c] | 12.59[d] |
| 95% CI | [7.32–8.72] | [7.66–10.39] | [10.23–10.75] | [11.31–12.26] | [10.91–12.54] | [12.43–12.75] |

**Notes:**
The overall mean is 11.83 (11.70; 11.96).
A different letter in superscript in the same row is indicative of a significant difference ($p < 0.05$), after the LSD *post-hoc* test; Rep. St., reproductive status; N/S, neutered/spayed; P.B., purebreed; C.-B., cross-bred; E, entire.

Significant differences were found between all the levels of the main factors analysed, except for cat types, where differences were not found significant between cross-bred and pure breed cats. Moggies live longer on average than 'pure breeds' and 'cross-bred', without significant differences being found between the last two. Relatively to 'sex', females have a longer average lifespan. Concerning 'reproductive status', 'neutered' tom cats, or 'spayed' queens have also significantly different lifespans, in comparison to 'entire' cats. The interaction between 'cat type' and 'reproductive status' showed that neutered/spayed cats have a longer average lifespan, while entire purebreds have the lowest average lifespan.

**Table 4 One-way analysis of variance (ANOVA) used to test significant differences in lifespan between 'reproductive status and sex' (entire male, entire female, neutered male, spayed female).**

| Source | Sum of squares | df | Mean square | F | p-value |
|---|---|---|---|---|---|
| Intercept | 781,932.661 | 1 | 781,932.66 | 24,345.40 | <0.001 |
| Factor | 11,828.907 | 3 | 3,942.969 | 122.764 | <0.001 |
| Corrected model | 11,828.907 | 3 | 3,942.969 | 122.764 | <0.001 |
| Error | 247,439.282 | 7,704 | 32.118 | | |
| Total | 1,337,278.71 | 7,708 | | | |
| Corrected total | 259,268.19 | 7,707 | | | |
| | Male, Entire | Female, Entire | | Male, Neutered | Female, Spayed |
| Mean | 9.40[a] | 10.89[b] | | 11.79[c] | 13.17[d] |
| 95% CI | [9.06–9.75] | [10.55–12.23] | | [11.59–12.00] | [12.95–13.80] |

Notes:
Mean lifespan (years), and respective 95% confidence intervals (CI) for the different levels of the factor.
$R^2 = 0.45$, Adjusted $R^2 = 0.45$; A different letter in superscript in the same row is indicative of a significant difference ($p < 0.05$), after the Tukey HSD *post-hoc* test.

**Table 5 Parameterization of the Cox regression modelling lifespan in cats, as a function of 'cat type' (moggy, cross-bred, cross breed), 'reproductive status' (entire, neutered/spayed), and 'sex' (male, female).**

| | $\beta$ | Wald | df | p-value | HR $e^{\beta}$ | $e^{\beta}$ 95% CI Lower | Upper |
|---|---|---|---|---|---|---|---|
| Sex, Male | 0.254 | 122.761 | 1 | <0.001 | 1.289 | 1.233 | 1.348 |
| Reproductive status, entire | 0.205 | 63.820 | 1 | <0.001 | 1.227 | 1.167 | 1.291 |
| Cat type | | 48.225 | 2 | <0.001 | | | |
| Cross-bred | −0.108 | 2.229 | 1 | 0.135 | 0.898 | 0.780 | 1.034 |
| Moggies | −0.251 | 44.951 | 1 | <0.001 | 0.778 | 0.723 | 0.837 |

Note:
The levels 'female', 'neutered/spayed', and 'pure breed' are used as reference in the model; HR, hazard ratio; CI, confidence interval.

The one-way ANOVA showed that spayed females have the longer average lifespan, followed by neutered males, entire females, and entire males. The results are presented in Table 4.

In both ANOVA tests, both the Levine's tests and the Kolmogorov-Smirnov tests showed homogeneity of variances ($p > 0.05$) and normal distribution of the residuals ($p > 0.05$).

The cox regression model was found significant (−2 Log Likelihood 122368, $\chi^2 = 222.69$, 4 df, $p < 0.001$). The full parameterization of the model, the respective statistical tests of parameters, and the hazard ratio (HR) for the parameters are shown in Table 5. The baseline function as covariate means values of: 'sex' 0.512, 'reproductive status' 0.273, 'cat type' pure breed 0, cross-bred 0.33, moggy 0.863. The plot of the baseline cumulative hazard is presented in Fig. 1. Figures 2–4 represent respectively the plots of the cumulative survivals for the three 'cat type', the two 'sex', and the two 'reproductive status'.
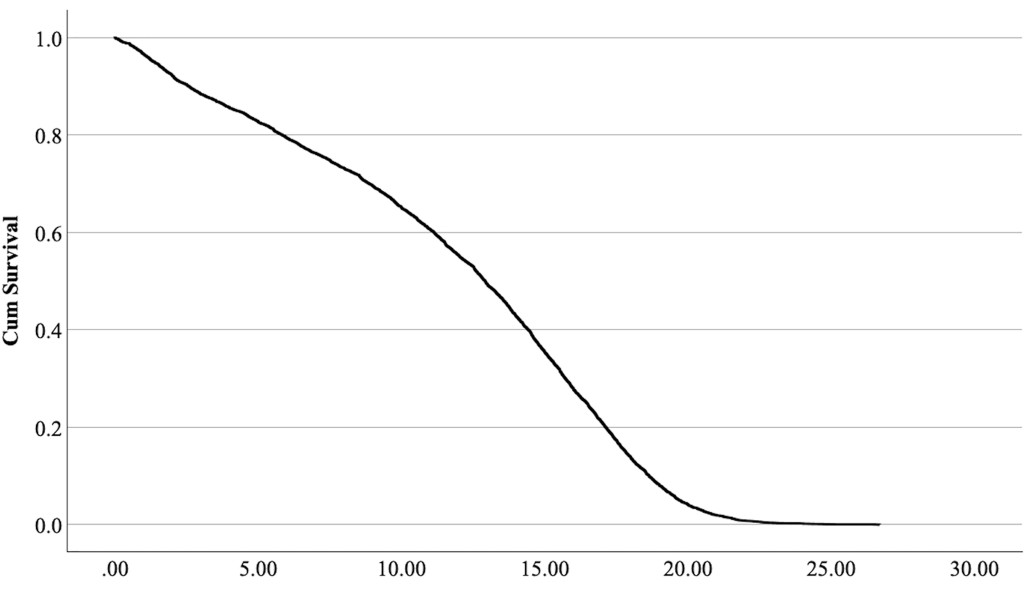

**Figure 1 Baseline cumulative survival function curve at means of covariates.**

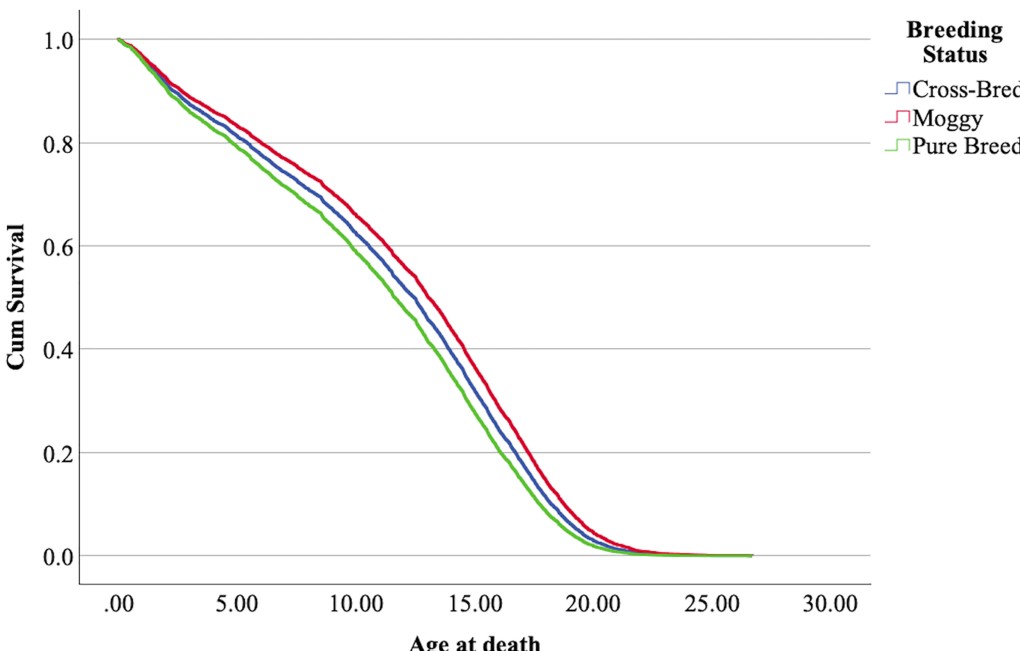

**Figure 2 Cumulative survival function curves for the three types of cats: pure breeds *vs* cross breed *vs* moggies.** The differences between cross-bred and pure breeds are not significant ($p > 0.05$).

The interpretation of the hazard ratio tells that 'males' had a lower probability of survival with an HR 28.9% higher than 'females'; 'entire' cats had a lower probability of survival with an HR 22.7% higher than 'neutered/spayed' cats; 'moggies' had a higher probability of survival with an HR 22.2% lower than both 'cross-bred' and 'pure-breed' cats, with no significant differences between these last two.

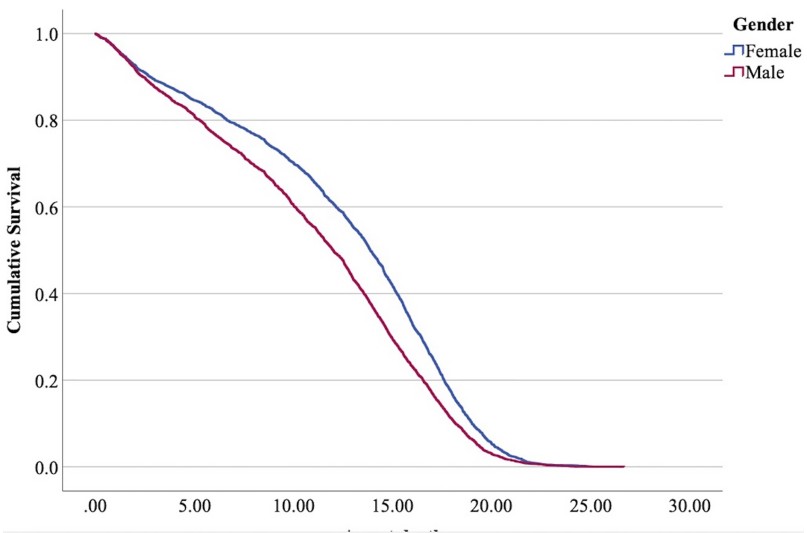

**Figure 3 Cumulative survival function curves for tom cats (males) *vs* queens (females).**

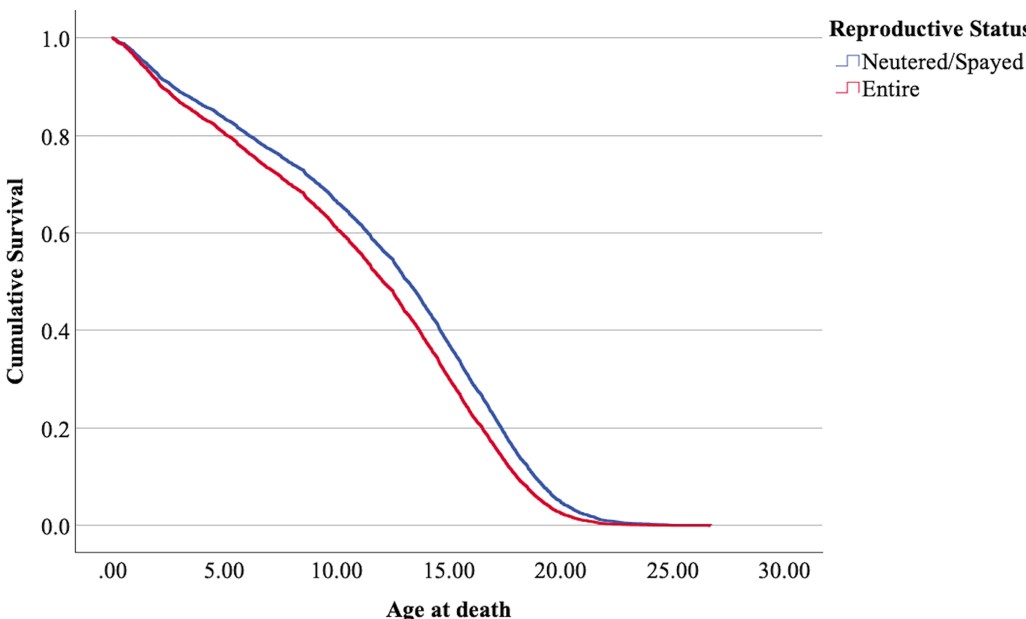

**Figure 4 Cumulative survival functions for neutered tom cats (males) and spayed queens (females), *vs* entire cats.**

The Kaplan-Meier model was found significant (Mantel-Cox log-rank 187335, $\chi^2 = 187.34$, 3 df, $p < 0.001$). The medians of the combined 'reproductive status' and 'sex', together with the respective confidence intervals, are shown in Table 6. The survival plot for the four combinations ('mollies', 'female entire', 'male neutered', and 'male entire') is presented in Fig. 5.

The means confidence intervals don't intersect each other and therefore were significantly different ($p < 0.05$). Nevertheless, the survival curves of neutered males and

**Table 6 Median lifespan (years) and respective interquartile range (IQR) for 'sex', 'cat type', 'reproductive status', and interaction between 'reproductive status', and 'sex'.**

| Factor | Level 1 | Level 2 | Level 3 | Level 4 |
|---|---|---|---|---|
| Sex | Male | Female | ——— | ——— |
| Median | 12.00 | 13.88 | ——— | ——— |
| IQR | 6.51; 15.69 | 8.60; 17.04 | ——— | ——— |
| Cat type | Moggy | Cross-bred | Pure breed | ——— |
| Median | 13.09 | 11.80 | 11.44 | ——— |
| IQR | 7.77; 16.59 | 5.90; 15.85 | 5.22; 15.32 | ——— |
| Reproductive status | Entire | Neutered/Spayed | ——— | ——— |
| Median | 11.22 | 13.37 | ——— | ——— |
| IQR | 3.25; 16.63 | 8.74; 15.87 | ——— | ——— |
| Sex × Reproductive S | Entire female | Entire male | Spayed female | Neutered male |
| median | 12.55 | 9.86 | 14.25 | 12.54 |
| IQR | 7.78; 16.67 | 2.90; 14.77 | 9.89; 17.16 | 3.53; 16.00 |

**Note:**
The overall median is 12.92 (7.44; 16.48).

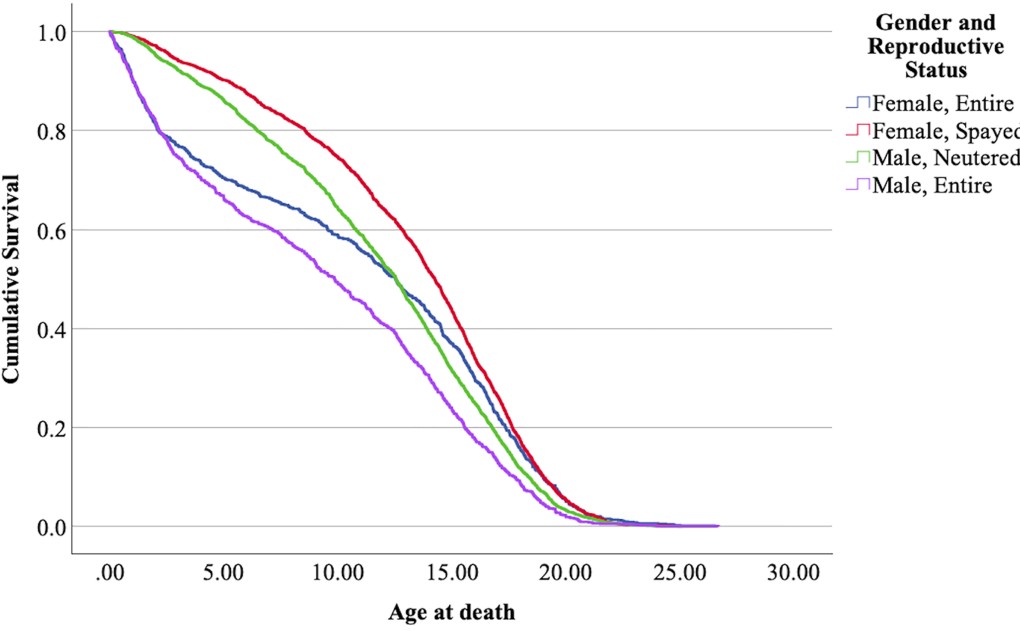

**Figure 5 Cumulative survival functions for neutered tom cats (males) *vs* spayed queens (females) *vs* entire tom cats (males) *vs* entire queens (females).**

entire females intersect each other at some point in time, with entire females showing a lower lifespan before reaching ≈ 12–13 years, but in later ages, lifespan increases comparatively. Mollies showed the longer lifespans, and entire males the lower.

## DISCUSSION

The overall mean lifespan, along with the respective 95% confidence interval, for the cats in the analyzed dataset was 11.83 years (11.70; 11.96). The median lifespan and the respective

**Table 7 Cats' lifespan estimation by sex.**

| Study | Location | Male cat | Female cat |
|---|---|---|---|
| Present study ($\theta$) | UK | 12.00 (6.51; 15.699) | 13.88 (8.60; 17.04) |
| Present study ($\mu$) | UK | 11.17 (10.99; 11.35) | 12.52 (12.33; 12.70) |
| O'Neill et al. (2015) ($\theta$) | UK | 13.0 (7.6; 16.0) | 15.0 (11.0; 17.4) |
| Montoya et al. (2023) ($\mu$) | USA | 10.72 (10.68; 10.75) | 11.68 (11.65; 11.71) |
| Gennaro, Isturiz & Pucheta (2023) ($\mu$) | Buenos aires | 10.92 (10.50; 11.35) | 12.26 (11.78; 12.73) |

Note:
Comparison of the results obtained in the present study with those obtained by other authors. Medians ($\theta$) with interquartile range and means ($\mu$) with 95% confidence intervals.

interquartile range were 12.92 years (12.73; 13.11). Other authors have reported median lifespans of 14.0 years (IQR 9.0–17.0; range 0.0–26.7) using UK data (O'Neill et al., 2015); 9.07 years (IQR 4.20–12.92; range 0.01–21.85) in a California, USA population (Kent et al., 2022); 11.18 years (mean 11.16–11.20) in a USA population (Montoya et al., 2023); 11.01 years (mean 10.55–11.46) in a population from Buenos Aires, Argentina (Gennaro, Isturiz & Pucheta, 2023); and over 12.5 years (median) in Sweden (Egenvall et al., 2009). The results of Egenvall et al. (2009) are close to those of the present study; however, the results of O'Neill et al. (2015) are higher, which can be explained by the fact that the authors excluded cats under five years of age from their analysis. The means obtained by Kent et al. (2022) are lower than those in the present study, likely due to local conditions affecting the cat population in California, as their results are also below those reported by Montoya and colleagues for the entire USA. The results obtained by Montoya et al. (2023) and Gennaro, Isturiz & Pucheta (2023) are lower than those in the present study, indicating higher survival rates in the UK. Nevertheless, it is important to note that different cat populations, influenced by varying environmental factors, exhibit different lifespans, as highlighted by Teng et al. (2024).

## The effect of sex

The effect of sex in the present analysis determined that male cats have a shorter life than female cats. Table 7 summarizes the results also in comparison with studies of other authors. All these authors confirm a higher lifespan for females.

Higher mortality rates for male cats may be attributed to traumatic, health-related, and genetic (epigenetic) factors. Traumatic causes include accidents and fights, which are more common among males than females. Agonistic behaviour between males is frequent for the definition of sexual dominance, which may result in serious injuries (Albright, Calder & Learn, 2022; Martin, 2023; Stella, 2021; Vitale, 2022) (Fig. 6). Tom cats, driven by their sexual instincts, roam over large areas in search of receptive females, while females typically do not venture out in search of males (Hall et al., 2016). O'Neill et al. (2015) identify trauma as the most common cause of mortality in England, accounting for 12.2%, with falls, traffic accidents, and dog attacks being notable contributors. Road traffic accidents are the leading cause of traumatic death for cats in both Sweden (Egenvall et al.,

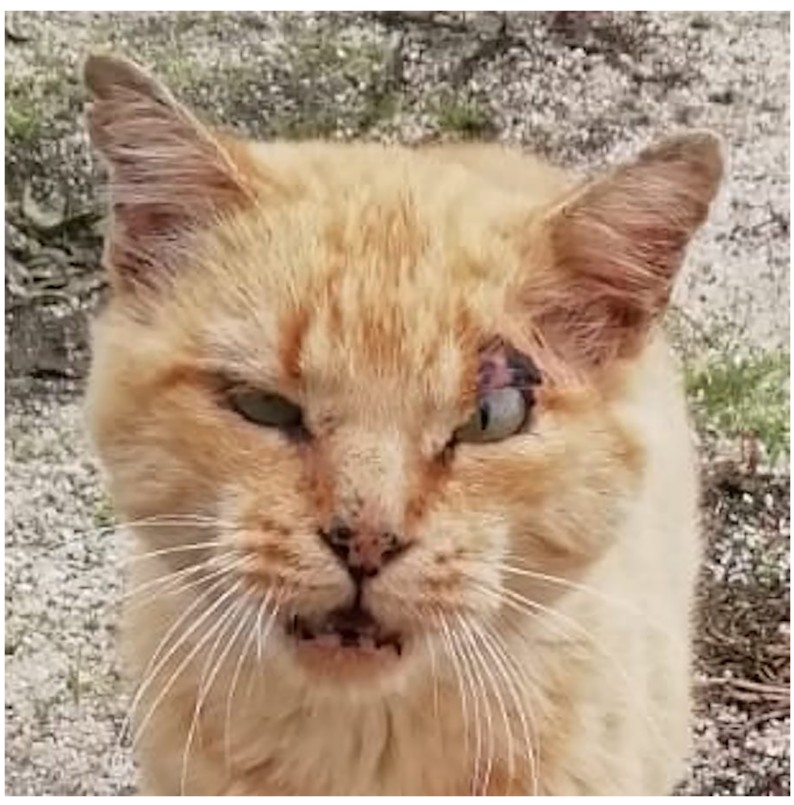

**Figure 6 A tomcat exhibiting his 'medals of honor'.** To note scars all over the face and the cropped ears.

*2009*) and the United States (*Loyd et al., 2013*), responsible for 60% to as much as 87% of traumatic deaths in cats (*Egenvall et al., 2009*; *Olsen & Allen, 2001*; *O'Neill et al., 2015*).

Several health conditions that are more prevalent in male cats can also shorten their lifespan. Urinary (urethral) tract obstruction has been identified as a condition predominantly affecting male cats (*Montoya et al., 2023*; *Segev et al., 2011*). Infections with feline immunodeficiency virus (FIV) and feline leukemia virus (FeLV) are also more common in males, as their tendency to roam outdoors increases their risk of exposure and infection (*Kent et al., 2022*). A study by *O'Neill et al. (2023a)* found that male cats have a higher predisposition to periodontal disease, renal tubular acidosis, heart murmurs, lameness, obesity, abscesses, wounds, and cat bite injuries (often related to fights between males). Periodontal disease in cats has been associated with comorbidities such as hepatic, cardiorespiratory, and renal disorders (*Watson, 1994*), as well as systemic bacteremia (*Perry & Tutt, 2015*).

The epigenetic likelihood of shorter lifespans in tom cats is highlighted by *Kent et al. (2022)*. This phenomenon is observed not only in cats but across various mammal species, as noted by *Clutton-Brock & Isvaran (2007)*. In polygynous species, more intense intrasexual male competition can limit opportunities for successful individual male breeding, resulting in weaker selection pressures favoring longevity in males compared to females.

**Table 8 Cats' lifespan estimation by reproductive status.**

| Study | Location | Neutered/Spayed | Entire |
|---|---|---|---|
| Present study ($\theta$) | UK | 13.37 (8.74; 15.87) | 11.22 (3.25; 16.63) |
| Present study ($\mu$) | UK | 12.45 (12.30; 12.60) | 11.83 (9.91; 10.40) |
| *O'Neill et al. (2014)* ($\theta$) | UK | 15.0 (11.8; 17.0) | 11.0 (2.13; 16) |
| *Gennaro, Isturiz & Pucheta (2023)* ($\mu$) | Buenos aires | 13.19 (13.14; 13.55) | 5.55 (5.48; 5.62) |

**Note:**
Comparison of the results obtained in the present study with those obtained by other authors. Medians (θ) with interquartile range and means (μ) with 95% confidence intervals.

## The effect of the reproductive status

The effect of reproductive status in the present analysis determined that neutered/spayed cats have a longer lifespan than entire cats. Other authors found values also agreeing with this tendency. Table 8 summarizes the results also in comparison with studies of other authors.

In agreement with the results of the present study, the authors of the studies in Table 6 confirm a higher lifespan for neutered/spayed cats. Other authors have considered reproductive status in conjunction with sex, which we will discuss in the following section.

Research has shown that neutered/spayed cats tend to have longer lifespans compared to those that are not (*Kraft, 1998*). This increased longevity in gonadectomized animals may result from a reduced likelihood of reproductive tract diseases and a decrease in risky behaviors (*Reichler, 2009*). *Gunther, Raz & Klement (2018)* found that a higher degree of emaciation due to disease and trauma was observed in entire free-roaming cats in Israel compared to those that were trapped, neutered/spayed, and returned. Additionally, the longer lifespan of neutered pets can be partially attributed to the heightened care that owners often provide (*Reichler, 2009*; *Dutton-Regester & Rand, 2024*). Due to the roaming behavior of entire tom cats, as previously explained, they are also more exposed to viral infections such as feline immunodeficiency virus (FIV) (*Mackintosh, 2024*) and are more likely to suffer traumatic death, especially at younger ages (*Gennaro, Isturiz & Pucheta, 2023*). Neutered males change their agonistic behaviour towards other males, reducing fighting and vocalizations, injuries, and transmission of infectious diseases resulting from fighting (*Finkler, Gunther & Terkel, 2011*).

## The effect of sex and reproductive status together

The effect of the sex and reproductive status in the present analysis determined that entire male cats (tom cats) have a shorter lifespan than all the other combinations, while spayed female cats (mollies) have a longer lifespan. In between, we have neutered male cats (gib cats) and entire female cats (queen cats). Entire females show an increased hazard ratio up to the age of ≈ 12.5 years (see Fig. 5), while from that age on, neutered males show, in comparison, a higher hazard ratio and lower life expectancy. Table 9 summarizes the results in comparison with studies of other authors.

The results obtained by *Kent et al. (2022)* in a Californian cat population are similar for mollies; however, the difference for entire females is enormous. This study, in comparison with the present one, also shows a very short lifespan for entire males, while the results for

**Table 9 Cats' lifespan estimation by sex and reproductive status.**

| Study | Entire female | Spayed female | Entire male | Neutered male |
|---|---|---|---|---|
| Present study ($\theta$) | 12.55 (7.78; 16.67) | 14.25 (9.89; 17.16) | 9.86 (2.90; 14.77) | 12.54 (3.53; 16.00) |
| Present study ($\mu$) | 10.87 (10.50; 11.30) | 12.52 (11.97; 13.36) | 9.40 (9.02; 9.79) | 11.79 (11.60; 11.99) |
| Kent et al. (2022) ($\theta$) | 4.68 (2.03; 10.36) | 10.48 (6.97; 13.68) | 3.67 (1.96; 8.70) | 9.84 (6.06; 13.04) |
| Gennaro, Isturiz & Pucheta (2023) ($\mu$) | 5.64[a] | 13.29[a] | 4.86[a] | 12.13[a] |

**Notes:**
Comparison of the results obtained in the present study with those obtained by other authors. Medians ($\theta$) with interquartile range and means ($\mu$) with 95% confidence intervals.
[a] Results reported without a confidence interval.

neutered males align with those of the present study. It is important to note that these results refer to cats that died at an age above one year, which further accentuates the differences. The results from *Gennaro, Isturiz & Pucheta (2023)*, shown without confidence intervals, reveal large disparities between entire and neutered/spayed cats. Interestingly, the graphical representation of life expectancy for the four combinations of sex and reproductive status over time shows the same pattern observed in the present study. In the earlier stages of life, entire females have a higher hazard ratio; however, at later stages (from approximately 12.5 years), the hazard ratio is higher for neutered males. This results in entire females having a higher life expectancy than neutered cats from around 12.5 years of age.

For all the reasons already discussed regarding sex and reproductive status, the results obtained for mollies and entire males were expected. The intriguing aspect of this interaction is the crossing of the survival functions for both entire females and neutered males. Without conclusive evidence, it is possible to speculate that in the early stages of female life, conceiving and giving birth increases the hazard ratio.

Some studies have identified an increased risk of developing prostate cancer at older ages in neutered dogs (*Bryan et al., 2007*). Nevertheless, prostate cancer has been documented in both entire and neutered cats, and no epidemiological studies have yet been conducted to find a predisposition (*Palmieri, Fonseca-Alves & Laufer-Amorim, 2022*). Additionally, male cats have been identified as being at greater risk of being overweight compared to female cats (*Courcier et al., 2012*; *Martins et al., 2023*; *McGreevy et al., 2008*). Overweight is also more frequent in neutered cats than in entire cats, peaking at around 9 years of age (*Martins et al., 2023*). Therefore, older neutered male cats are at a higher risk of overweight and obesity, which have been identified as comorbid factors in chronic diseases in cats (*Corbee, 2014*; *Teng et al., 2018*).

The analysis of the influence of reproductive status on cats' lifespan needs to be considered carefully, as the variability in the age of neutering/spaying is not accounted for in the present study.

## The effect of cat type

The effect of cat type in the present analysis determined that moggies have a longer lifespan than pure breed and cross-bred. Table 10 summarizes the results also in comparison with studies of other authors.

**Table 10 Cats' lifespan estimation by cat type.**

| Study | Location | Moggies | Cross-bred | Pure breed |
|---|---|---|---|---|
| Present study ($\theta$) | UK[a] | 13.09 (7.77; 16.59) | 11.80 (5.90; 15.85) | 11.44 (5.22; 15.32) |
| Present study ($\mu$) | UK[a] | 12.01 (11.87; 12.15) | 11.03 (10.32; 11.74) | 10.56 (10.16; 10.96) |
| *Montoya et al. (2023)* males ($\mu$) | USA[b] | | 11.98 (11.91; 12.06) | 11.05 (10.97; 11.14) |
| *Montoya et al. (2023)* females ($\mu$) | USA[b] | | 11.62 (11.58; 11.65) | 10.66 (10.63; 10.7) |
| *O'Neill et al. (2014)* ($\theta$) | UK[c] | | 14.0 (9.1; 17.0) | 12.5 (6.1; 16.4) |
| *Gennaro, Isturiz & Pucheta (2023)* ($\mu$) | Buenos aires[d] | | 10.97 (10.81; 11.13) | 14.81 (14.51; 15.10) |

**Notes:**
Comparison of the results obtained in the present study with those obtained by other authors. Medians ($\theta$) with interquartile range and means ($\mu$) with 95% confidence intervals.
[a] Despite a tendency for a higher lifespan of cross-bred in comparison to pure breed cats, these are results non-significative.
[b] This study has separate data for males and females and means for cross-bred and moggies are calculated together.
[c] In this study means for cross-bred and moggies are calculated together.
[d] In this study means for cross-bred and moggies are calculated together.

Never before has a study separated the breeding status of cats into the three groups defined in the present study (moggies, cross-bred, and purebred). Previous studies have typically considered only two groups: purebreds and all others, combining moggies (those with no purebred ancestors) with cross-breds (defined in the present study as having only one purebred ancestor). All previous studies found significant differences in cats' lifespans between purebreds and other types, a finding reiterated by the results of the present study. Despite the lack of statistical significance, we observe a trend toward an increased lifespan in the group defined as cross-bred compared to purebreds. These results suggest that increasing genetic diversity contributes to greater longevity.

It is recognized that purebred animals, due to a higher proportion of homozygous genes, have a greater probability of expressing recessive disorders caused by deleterious genes (*Doekes et al., 2019*; *Fernández et al., 2002*; *Mata & Mata, 2023*). This phenomenon is known as inbreeding depression, resulting from a higher inbreeding coefficient (*Charlesworth & Charlesworth, 1999*).

Certain pure breeds are known to be prone to specific hereditary conditions, such as polycystic kidney disease (*Michel-Regalado et al., 2022*), hypertrophic cardiomyopathy (*Korobova & Kruglova, 2024*), hyperthyroidism (*Mata & Bhuller, 2022*), and respiratory issues associated with the brachycephalic anatomy of certain breeds, like the Persian (*Farnworth et al., 2016*). In contrast, moggies, with their more diverse gene pool, are generally thought to be healthier and more resilient.

In comparison to dogs, where 69.4% were purebred in 2019 (*O'Neill et al., 2023b*), the purebred cat population in the UK is much lower, at 12% in 2019 (*O'Neill et al., 2023a*). This fact may contribute to a less-defined difference between the three types of cats considered. In a similar study with dogs (*Mata & Mata, 2023*), significant differences between cross-bred and purebred dogs (with a higher hazard ratio) were well established.

## CONCLUSIONS

This study emphasizes the importance of considering multiple factors when assessing cat longevity, suggesting that owners should be mindful of 'sex', 'reproductive status', and

breeding status or 'cat type' when caring for their pets. The observed variations in lifespans across different populations and environmental contexts further indicate the necessity for tailored approaches to feline health and welfare, such as giving preference to outbreeding practices. Future research should explore the underlying mechanisms influencing these trends to develop targeted interventions aimed at enhancing the quality of the health and welfare, and duration of 'cats' lives.

### Funding

The Foundation for Science and Technology (FCT, Portugal) provided financial support *via* CISAS UIDB/05937/2020 (DOI: 10.54499/UIDB/05937/2020) and UIDP/05937/2020 (DOI: 10.54499/UIDP/05937/2020) including the contract of the author. The funders had no role in study design, data collection and analysis, decision to publish, or preparation of the manuscript.

### Grant Disclosures

The following grant information was disclosed by the authors:
The Foundation for Science and Technology (FCT, Portugal) provided financial support *via* CISAS: UIDB/05937/2020, UIDP/05937/2020.

### Competing Interests

Fernando Mata is an Academic Editor for PeerJ.

### Author Contributions

- Fernando Mata conceived and designed the experiments, performed the experiments, analyzed the data, prepared figures and/or tables, authored or reviewed drafts of the article, and approved the final draft.

### Data Availability

The data is available at figshare: O'Neill, Dan; Teng, Kendy Tzu-yun (2024). Life tables of annual life expectancy and risk factors for mortality in cats in the United Kingdom. figshare. Dataset. https://doi.org/10.6084/m9.figshare.25006802.v1.

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
