# Peer review of "Life expectancy of cats in Britain: moggies and mollies live longer"

_PeerJ, doi:10.7717/peerj.18869_

## Round 0.1 · original submission · Major Revisions

Dear Dr. Mata, I kindly request you to pay attention to the reviewers' comments and make changes and additions to this manuscript. I hope that this will enable the reviewers to approve the publication of this article.

Reviewer 1 ·

Basic reporting

Comments to the Author

Comments:
Manuscript ID: -108700, Title: “Life expectancy of cats in Britain: moggies and mollies live longer”

The present study has original information on the longevity of cats in Britain. Although the study is interesting, I make a series of comments and suggestions that I hope will be useful to improve the manuscript. Therefore, I invite the author to review and answer all the questions and comments that I share below.

In the title, why include "mollies"? On the other hand, if it goes in the title, it should be described throughout the work.

An interesting and novel part of the article is about Moggies cats (as the author mentions in L:279-280), however, in the introduction nothing is said about these cats. Therefore, I consider it important that the author develops a few sentences or a short paragraph about these cats, especially for those who are not from the UK.

Be very careful with the verb tenses, they are written in the past and in several parts of the manuscript they are written in the present, which should be changed: L128, 169, etc..

In the writing section, it is striking that the work is the work of only one person, but nevertheless, in several parts of the manuscript, the plural is used, for example L32, 186, 178, 218, 238, 284, etc.

Experimental design

Regarding the cat data, from what calendar year range was it evaluated? How many decades back?

L102: Why did the author use LSD? He should explain very clearly why he used this test (LSD) of multiple comparisons, and not another one that fits better, such as Bonferroni or Tukey?

L105-107: Did all the data have a normal distribution? Besides the QQ plot, did you use any tests to check for normality, or to find the appropriate distributions for each variable?

L109: What is the difference between "moggy" and "crossbred"?

L110: The term "gender" is used for humans, for non-human animals the correct term is biological sex.
What level of significance was used?

Validity of the findings

L139-141: According to the results shown in Table 2, there is no significant interaction between gender and reproductive status (p=0.447), therefore the different genders cannot be compared with the different reproductive statuses among them. This should be modified.

L176: To say that "are significantly higher" you have to use a testing method, which one was used to compare the data between the two studies? Please avoid using inappropriate terminology to show or discuss descriptive results. The same occurs in other parts of the manuscript, such as L178, 181, etc.

L185: change gender for sex

Additional comments

On the other hand, it is interesting to note that this work is presented by an author from Portugal (sole author), but with data from the United Kingdom. Can the author provide a contextual framework for this study?

L189-200: In this particular section, but also throughout the manuscript, I wonder why it was not included whether the cats are indoor or outdoor? Can the author explain a little more about this information. Because a lot of information can be biased in relation to this element that is fundamental in relation to the longevity of cats.
For example, is there a difference in the sex of cats that are kept indoors or outdoors?

L217: Similar question. Is there any relationship between neutered cats and less outdoor exposure and more care?

L237-273: This entire paragraph needs to be restructured, given as I mentioned before, and the author himself showed, there was no significant interaction between gender and reproductive status.

L273: This sentence should be further developed and explained by the author. What does he mean by that?

L279-280: Why does the author think cats have never been separated into those 3 groups by breed status?

L304-305: This sentence is not a conclusion that can be drawn from this article.

The figures are of very poor quality, it is suggested to improve them.

·

Basic reporting

The purpose of this manuscript is clear. The value of the manuscript is sound.

a. General:
i. Felis catus should be italicized.
ii. The author should use sex instead of gender.
iii. The use of present and past tense in this manuscript is inconsistent. For example, on line 24: “were” implies cats used to be kept in household for hunting in the past. Yet, the author cited a reference in 2020. The review suggests using present tense instead.
iv. The writing of this manuscript needs some work. Some sentences are too long and convey too many ideas at once, making the logic of the ideas unclear. Many sentences are vague, and the logic of the sentences is unclear. The review suggests a major revision of the language of the manuscript. Please revisit the entire manuscript, the specific point below are examples only.
b. Abstract:
i. Line 19: This sentence is unclear. How does selective breeding result in breeds defined by appearance? This sentence implies that cats have breeds defined by appearance but not dogs.
ii. Line 24, 26, 32: Animals do not have genders.
c. Introduction:
i. Line 44-45: It’s unclear what “a process of self-domestication in a commensal relationship” mean. Please revise and provide examples.
ii. Line 47-48: It’s unclear about the relevance of this sentence to the goal of this study. It seems vague and everything can provide enjoyment. It’s unclear how it is relevant to role specialization in the next sentence. Dogs have specialized jobs not because they provide enjoyment, affection, companionship, or health benefits, they developed into specific roles because they can perform tasks.

Experimental design

Materials and methods
i. Line 102: post-hoc
ii. Line 120: I don’t think in the current age, reporting that data were input into excel is necessary. I recommend removing this sentence entirely. However, the following SPSS program should be reported.
iii. Line 94: Please clarify what it means by "they had at least one electronic patient record in that year." Is the author saying that the clinic that affiliated with the program had at least one patient record? If the clinic did not have any patient record, why would it be in this study at all. Is the author saying that the patient in the clinic had at least one record in the system? How did the author(s) handle the records when one patient have multiple records?

iv. Line 95: You have age at death as one of the variables. How about date of birth? What happens when you don't have date of birth but you have age at death? What happens if the cats are alive?
v. Line 100: define outliers

Validity of the findings

General: Need to see revised materials and methods before evaluating the results. Results need to be rewritten as well. The statistical method is sound.
For example:
Line 128: was 10.605 years. consider rounding to nearest 100th
Line 130: it's either significant or not significant. Highly significant is not statistically
Line 165: longer, not larger lifespan

Table 2, 3, and 5: In table 3 post-hoc test, the author(s) tested gender x reproductive status. However, in table 2, the p-value for gender x reproductive status is 0.447. Shouldn't the author test reproductive status x breeding status in the post-hoc test instead?

---

## Round 0.2 · Major Revisions

I ask you to conduct statistical analysis of the data very carefully in accordance with the comments of the reviewers. The journal will not be able to publish unreliable results. I hope that the new version of your manuscript can be recommended for publication by the reviewers. Please be very careful.

Reviewer 1 ·

Basic reporting

Ok

Experimental design

I thank the author for responding to the questions and comments made.

However, with respect to this statement "The prerequisites of the ANOVA are met with the normal distribution of the residuals (not the variables). Therefore I have checked normal distribution of residuals only. As the ANOVA is a robust test accepting small deviations from normality in the residuals, a Q-Q plot check only was made. No normality tests implemented.",

What do you mean by "small deviations"?

Although the QQ-plot is used, it is very subjective. Therefore, I request that the author perform the distribution of the residuals to corroborate that they follow a normal distribution using a test. I also request that he share with us the QQ-Plot of each variable and ANOVA as well as the p values ​​of the corresponding Tests to verify the distribution in each case.

Validity of the findings

Ok

Additional comments

Ok

·

Basic reporting

Felis catus is still not italicized in the R1 manuscript version.

L149, 152, 154, 180, 235, 256: the text still says larger lifespan instead of longer lifespan. Please recheck.

L43: is it really true that cats are primarily indoor for rodent control? In the past, cats are kept in barns and ships for rodent control. I could see that being "indoor". Many cats (in my neighborhood) are outdoor.

L47: can you define health benefits?

L113: Tukey's HSD instead of Tuckey's.

L156-158: fragmented sentence. "The baseline functions..."

Minor comments:
L140: delete space before "The ANOVA"

Experimental design

Since there were no personal information release, I suspect that prior permission or confidentiality agreements were not necessary. If there was, would you mind elaborating on the data usage policy in your study?

Validity of the findings

Statistical tests are sound. Data are clearly presented without over-reaching stipulations or conclusions.

---

## Round 0.3 · accepted · Accept

Dear authors, I congratulate you on the acceptance of this article for publication.

Reviewer 1 ·

Basic reporting

ok

Experimental design

I think it's good that the author included a test to assess the normality of the residuals. I think that in this case Shapiro-Wilks would have been better, but there is no problem in leaving the analysis with Kolmogrov-Smirnov.

Validity of the findings

ok

Additional comments

ok